# Cryo-EM elucidates the uroplakin complex structure within liquid-crystalline lipids in the porcine urothelial membrane

Haruaki Yanagisawa[1], Yoshihiro Kita[2,3], Toshiyuki Oda [4✉] & Masahide Kikkawa [1✉]

The urothelium, a distinct epithelial tissue lining the urinary tract, serves as an essential component in preserving urinary tract integrity and thwarting infections. The asymmetric unit membrane (AUM), primarily composed of the uroplakin complex, constitutes a critical permeability barrier in fulfilling this role. However, the molecular architectures of both the AUM and the uroplakin complex have remained enigmatic due to the paucity of high-resolution structural data. In this study, we utilized cryo-electron microscopy to elucidate the three-dimensional structure of the uroplakin complex within the porcine AUM. While the global resolution achieved was 3.5 Å, we acknowledge that due to orientation bias, the resolution in the vertical direction was determined to be 6.3 Å. Our findings unveiled that the uroplakin complexes are situated within hexagonally arranged crystalline lipid membrane domains, rich in hexosylceramides. Moreover, our research rectifies a misconception in a previous model by confirming the existence of a domain initially believed to be absent, and pinpointing the accurate location of a crucial *Escherichia coli* binding site implicated in urinary tract infections. These discoveries offer valuable insights into the molecular underpinnings governing the permeability barrier function of the urothelium and the orchestrated lipid phase formation within the plasma membrane.

[1] Department of Cell Biology and Anatomy, Graduate School of Medicine, The University of Tokyo, 7-3-1 Hongo, Bunkyo-ku, Tokyo 113-0033, Japan. [2] Life Sciences Core Facility, Graduate School of Medicine, The University of Tokyo, 7-3-1 Hongo, Bunkyo-ku, Tokyo 113-0033, Japan. [3] Department of Lipidomics, Graduate School of Medicine, The University of Tokyo, 7-3-1 Hongo, Bunkyo-ku, Tokyo 113-0033, Japan. [4] Department of Anatomy and Structural Biology, Graduate School of Medicine, University of Yamanashi, 1110 Shimokato, Chuo, Yamanashi 409-3898, Japan. ✉email: toda@yamanashi.ac.jp; mkikkawa@m.u-tokyo.ac.jp

The urothelium, which lines the urinary tract, performs a dual function: it serves as a permeability barrier, effectively preventing the leakage of urine components into surrounding tissue while undergoing morphological changes to adapt to the distention and contraction that occurs during the micturition cycle[1–4]. The apical surface of the urothelium is covered by numerous structurally rigid membrane plaques, known as asymmetric unit membranes (AUM), as observed by electron microscopy[4–6]. The AUMs are composed of hexagonally arranged 16 nm uroplakin complexes containing four major proteins, UPIa, UPIb, UPII, and UPIIIa[7]. UPIa and UPIb have four transmembrane domains and belong to the tetraspanin family[8]. Meanwhile, UPII and UPIIIa each possess a single transmembrane domain and form Ia/II and Ib/IIIa heterodimers within the endoplasmic reticulum (ER) in conjunction with UPIa and UPIb, respectively[9,10]. In the post-Golgi compartment, UP heterodimers are assembled into paracrystalline arrays[11]. The AUM's quasi-crystalline symmetry has been leveraged in the structural investigation of the uroplakin complex, as documented by several studies[12–16], the resolution of the resulting electron density map remains insufficient for a comprehensive study of the secondary structures of the extracellular domains or for the localization of individual UP subunits.

The uroplakin complex also plays a crucial role in the development of urinary tract infections, as it facilitates *Escherichia coli*'s attachment to the urothelium[17,18]. This attachment is facilitated by the interaction between the N-glycosylated UPIa's mannose residues and the FimH lectin located at the tip of the bacterium's pili[19]. The binding of FimH to UPIa triggers phosphorylation of UPIIIa and subsequent apoptosis of the urothelial cells[20]. Furthermore, it has been postulated that this FimH-UPIa interaction induces conformational alterations in the transmembrane helix bundles[16]. However, it is unclear whether FimH's binding to a flexible carbohydrate chain can change the conformation of a rigid uroplakin complex. In addition, the positioning of the heterodimer has been inferred by comparing the uroplakin structure with and without bound FimH[14]. Nevertheless, the low resolution of the electron density map hinders the validity of these conclusions.

In this study, we reconstructed the 3D structure of the uroplakin complex by observing tilted AUMs using cryo-electron microscopy. By combining random conical tilt and single particle analysis[21–24], we were able to build an accurate model of the uroplakin hexameric complex.

## Results

**Cryo-electron microscopy of AUM.** To determine the high-resolution structure of the uroplakin complex, we isolated the AUM from porcine urinary bladders, leveraging its resistance to sarkosyl[25], and employed cryo-electron microscopy techniques. Initially, we used cryo-electron tomography to observe the AUM; however, the obtained structure was of low resolution as a result of the irradiation-induced distortion of the specimen[26,27]. Subsequently, we performed single particle analysis on AUM samples tilted up to 55° (Fig. 1a, b), resulting in a global resolution of 3.5 Å, with 3.2 Å and 6.3 Å resolutions in the transverse and vertical directions, respectively (Figs. 1c, 1d, S1a–d). These resolutions allowed us to construct a model of the main chains and some side chains (Fig. S1e, f), based on AlphaFold predictions, with the exceptions of the 90–101 loop of UPII and the cytoplasmic loop of UPIIIa due to their flexibility[28] (Fig. 2).

To our surprise, we found that the hexagonally arranged liquid-crystalline lipids were not only within the central pore but also in the inter-complex regions, spanning the entire outer leaflet of the AUM (Figs. 1d, 2, Supplemental Movie 1). The crystalline

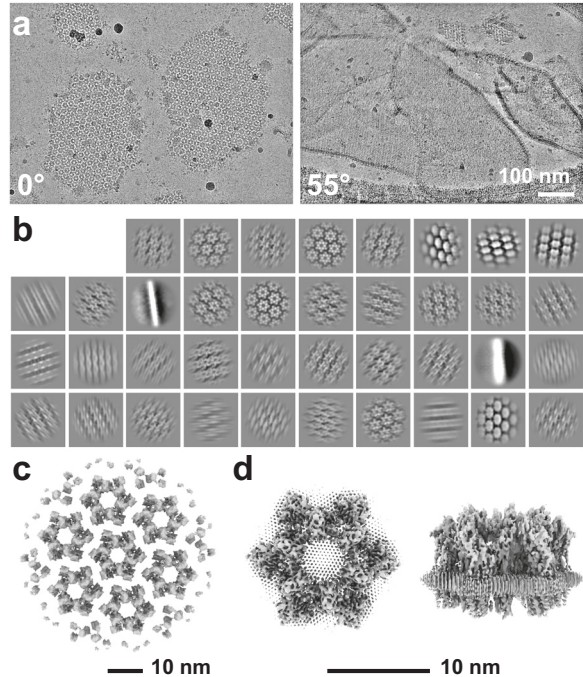

**Fig. 1 Cryo-electron microscopy of the AUM. a** Representative micrographs of the AUM. Tilt angles are indicated. **b** 2D class averages of the particles. **c** Reconstructed hexagonal array of the uroplakin complex. **d** Local refinement of the central uroplakin complex. Top and side views are shown. Hexagonally-aligned crystalline lipids are visualized.

lipids were tightly packed, with a distance of ~4.8 Å between them[29,30]. Further local refinement focusing on the extracellular domain revealed the additional outermost part of the lipid densities, suggesting a structural heterogeneity in the association between the lipids and the uroplakin complex (Fig. S1a). Some lipids appeared to be stabilized by vertically-aligned bulky side chains (Fig. 3)[31]. Although the majority of ordered sphingolipids are typically found in the outer leaflet of the plasma membrane due to the asymmetric localization of cholesterol[32–34], weak signals of inner leaflet lipids were detected between the UPIa and Ib transmembrane domains (Fig. S2a). The presence of hexagonally-aligned lipids in the inner leaflet possibly contributes to the AUM's exceptional rigidity. These crystalline lipids in the AUM are likely to play an essential role in the barrier function of the urothelium (see Discussions).

**Subunit arrangement of the uroplakin complex.** The cryo-EM structure of the uroplakin complex reveals that the extracellular domains of both UPII and UPIIIa exhibit distinctive beta-sheet structures (Fig. 2b), which could have functional implications in protein binding or adhesion of pathogens. With these extracellular domains of UPII and UPIIIa, uroplakin heterotetramer overall adopts a Y-shaped conformation, which creates a substantial groove that may also provide a binding interface for other proteins (Fig. S2b, white arrow). Adjacent heterotetramers are connected through arches formed by the UPII and UPIIIa extracellular domains. Interestingly, beneath these arches, a channel is present, which is continuous with the central cavity space (Fig. S2b, red arrows). The biochemical robustness of the uroplakin hexameric complex, despite its abundance of inner cavities, can likely be attributed to the reinforcement provided by the lipid domains.

The prevailing hypothesis concerning the subunit arrangement of the uroplakin complex suggests that Ia/II and Ib/IIIa

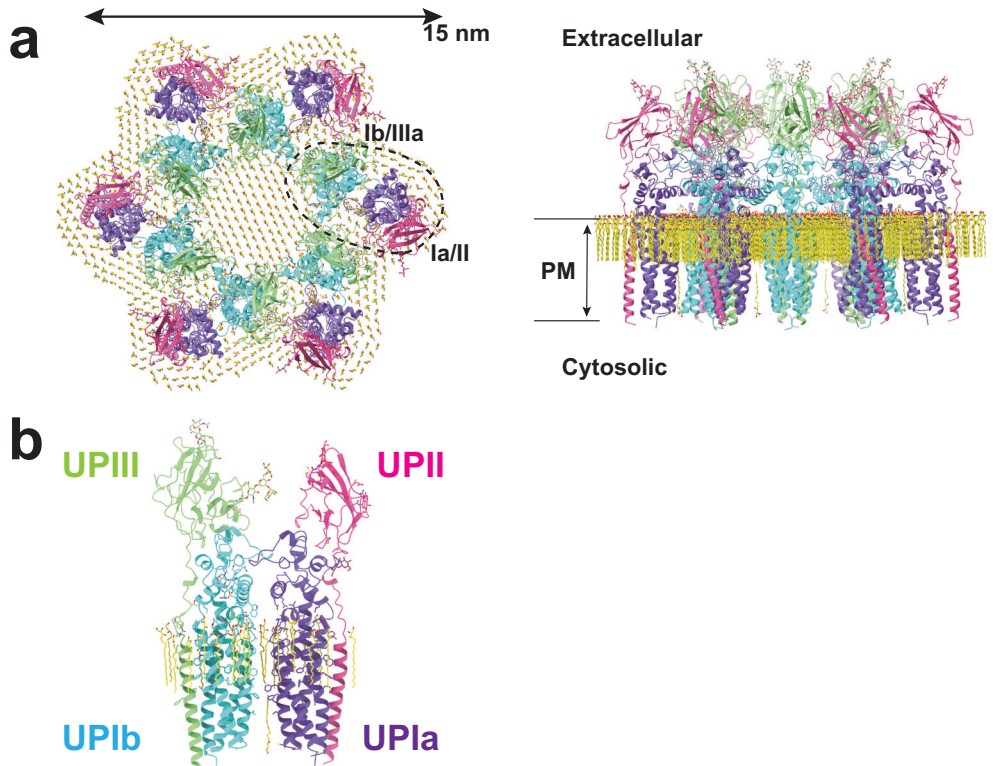

**Fig. 2 Models of the uroplakin complex and the crystalline lipids.** Models of the hexameric complex (**a**) and single heterotetramer (**b**) are shown. An initial model was predicted using AlphaFold and real-space refined using PHENIX and ISOLDE. Lipid models (C16 ceramide or sphingosine) were manually fitted and refined using ISOLDE. Note that the models of sphingolipids and ceramides are placed tentatively. The precise identities of these lipids within the structure remain uncertain. Purple: UPIa; cyan: UPIb; pink: UPII; green: UPIIIa; and yellow: lipids.

heterodimers form the inner and outer subdomains, as inferred from electron microscopy studies of FimH-bound AUM[14,19]. However, our high-resolution reconstruction revealed that Ia/II and Ib/IIIa heterodimers constitute the outer and inner subdomains, respectively (Fig. 2a). Additionally, it has been reported that the cleavage of UPII at Arg84 by furin results in the removal of the N-terminal prosequence (amino acids 25-84) from the uroplakin complex[9,35]. However, our findings suggest that the prosequence forms the external surface of the complex through the formation of several beta sheets (Fig. S2c). Moreover, our reconstruction uncovered the presence of carbohydrate chains on six N-glycosylated residues, namely UPIa Asn169, UPIb Asn131, UPII Asn28, Asn57, UPIIIa Asn139, and Asn170 in accordance with previous studies (Fig S3)[9,36,37]. Previous electron microscopy studies have suggested that UPIa Asn169, functioning as the binding site for the bacterial FimH protein, is located at the apex of the inner subdomain[14,38]. However, our reconstruction placed this glycosylated residue on the external surface of the uroplakin complex. Our structural analysis of the FimH-bound uroplakin complex also supports this assignment (see *Structure of FimH-bound uroplakin complex* section below).

**Inter-subunit interactions within the uroplakin complex**. Our map and model suggest that the hexameric complex is assembled through three inter-subunit interfaces: the Ia-Ib, II-IIIa, and IIIa-IIIa interfaces. The Ia-Ib interface appears similar to that reported in the PRPH2-ROM1 tetraspanin complex and consists of two opposing helices (Fig. 4a)[39]. The resolution of the II-IIIa interface, particularly near the flexible loop 90–101 of UPII, is of lower quality. However, it appears to be stabilized by a hydrophobic cluster of a disulfide bond between Cys51-80 and Pro79 of UPII, a disulfide bond between Cys47-Cys110, and Leu113 of UPIIIa

(Fig. 4b). The UPIIIa extends its loop 137–161 into the groove of an adjacent IIIa, forming an interface stabilized by a hydrophobic cluster composed of Pro146 of IIIa, Leu66, and Trp182 of the adjacent IIIa (Fig. 4c). Additionally, a hydrophobic cluster between the transmembrane helices of UPIa and Ib appears to stabilize the heterotetramer (Fig. 4d). These multiple interactions are thought to contribute to the highly rigid nature of the uroplakin complex and the AUM.

**Lipidomic analysis of the AUM**. A lipidomic analysis was conducted to discern the lipid species comprising the characteristic paracrystalline array of the AUM. The lipid composition of the sarkosyl-insoluble fraction, primarily consisting of the AUM, was compared to that of the total urothelial membrane homogenates (Fig. 5 and Supplemental Data 2). The results, as depicted by the LC–MS profile, revealed the enrichment of hexosylceramides in the AUM, in accordance with the previous studies[40–42]. We also observed enrichment of ceramides, which is likely to be fragmentation products of ionization. These results suggest that the hexagonally arranged lipid domain is composed of hexosylceramides.

**Structure of FimH-bound uroplakin complex**. We have solved the structure of the uroplakin complex in association with FimH (Fig. 6)[16]. Binding of FimH to the AUM was confirmed by SDS-PAGE and fluorescence microscopy (Fig. S4a, b). Initially, the FimH-bound map displayed a lower resolution and weaker lipid densities compared to the FimH-unbound map (Fig. S4c, pre-3D classification). To enhance resolution and clarify the lipid array, we employed 3D classification, resulting in a higher resolution map (Figs. 6a and S4c, class #1). However, no marked structural differences were discernible between the FimH-bound and

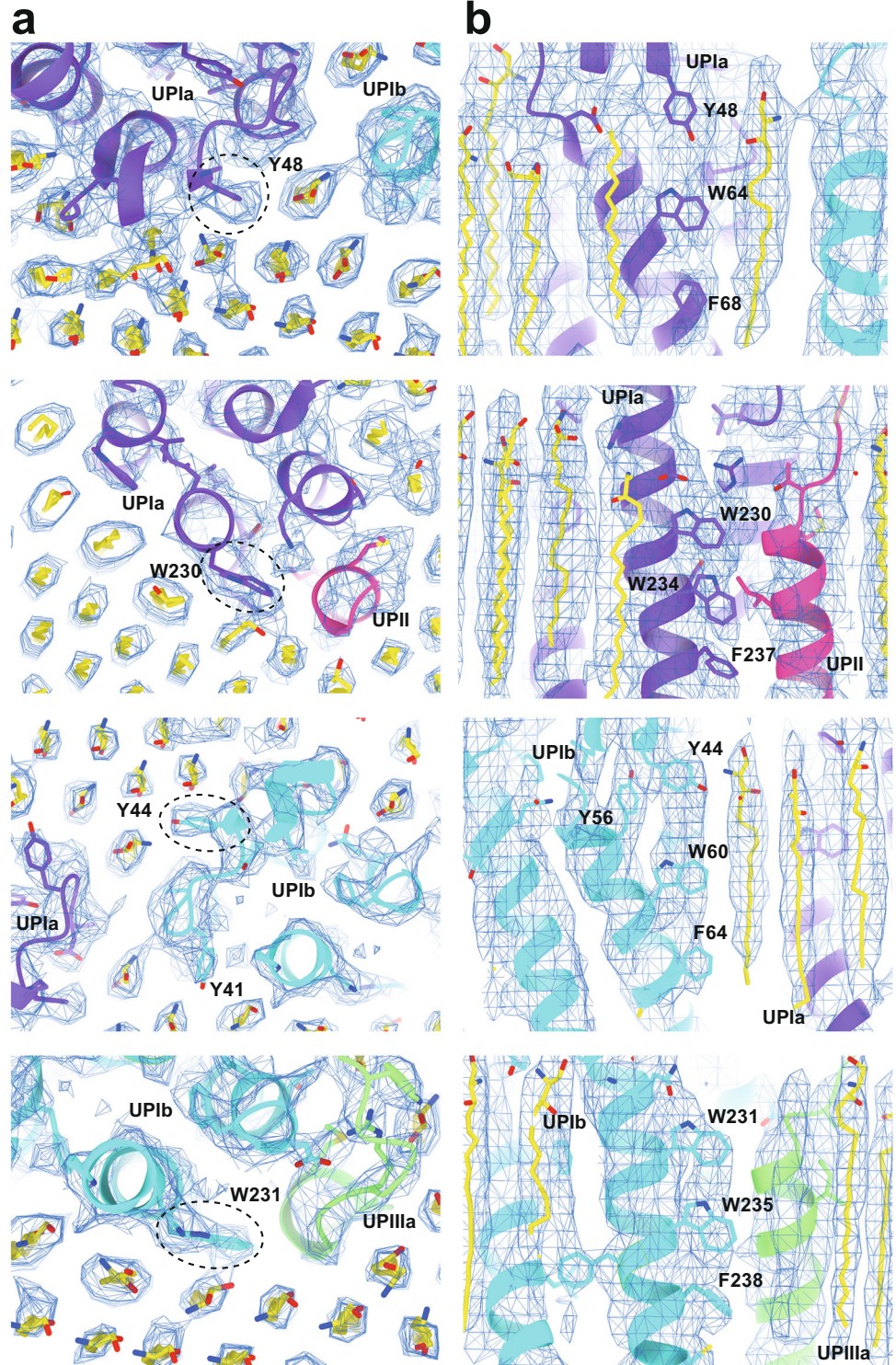

**Fig. 3 Interactions between the UPIa/Ib transmembrane helices and the lipids.** Vertically-aligned bulky side chains appear to be incorporated into the crystalline lipids. Top (**a**) and side (**b**) views are shown.

unbound maps (Fig. 6b). In the lower resolution class #2 and #3 maps, lacking the hexagonal crystalline lipid array, we identified additional densities adjacent to UPIa Asn169 (Fig. 6c, arrowheads). These are hypothesized to represent bound FimH components and/or FimH-stabilized carbohydrate chains. Our findings suggest that FimH binding may modify the uroplakin-lipid array arrangement without affecting the uroplakin complex conformation.

## Discussion

The structures of the AUM and the uroplakin have been extensively analyzed utilizing electron microscopy[12–15,19]. One of the studies achieved a resolution of approximately 6 Å in the membrane plane, yet the vertical resolution was limited to 12.5 Å, thus failing to uncover the residue-level information[12]. This limited structural information led to incorrect subunit assignment of the uroplakin[14]. The Ia/II heterodimer has long been believed to form

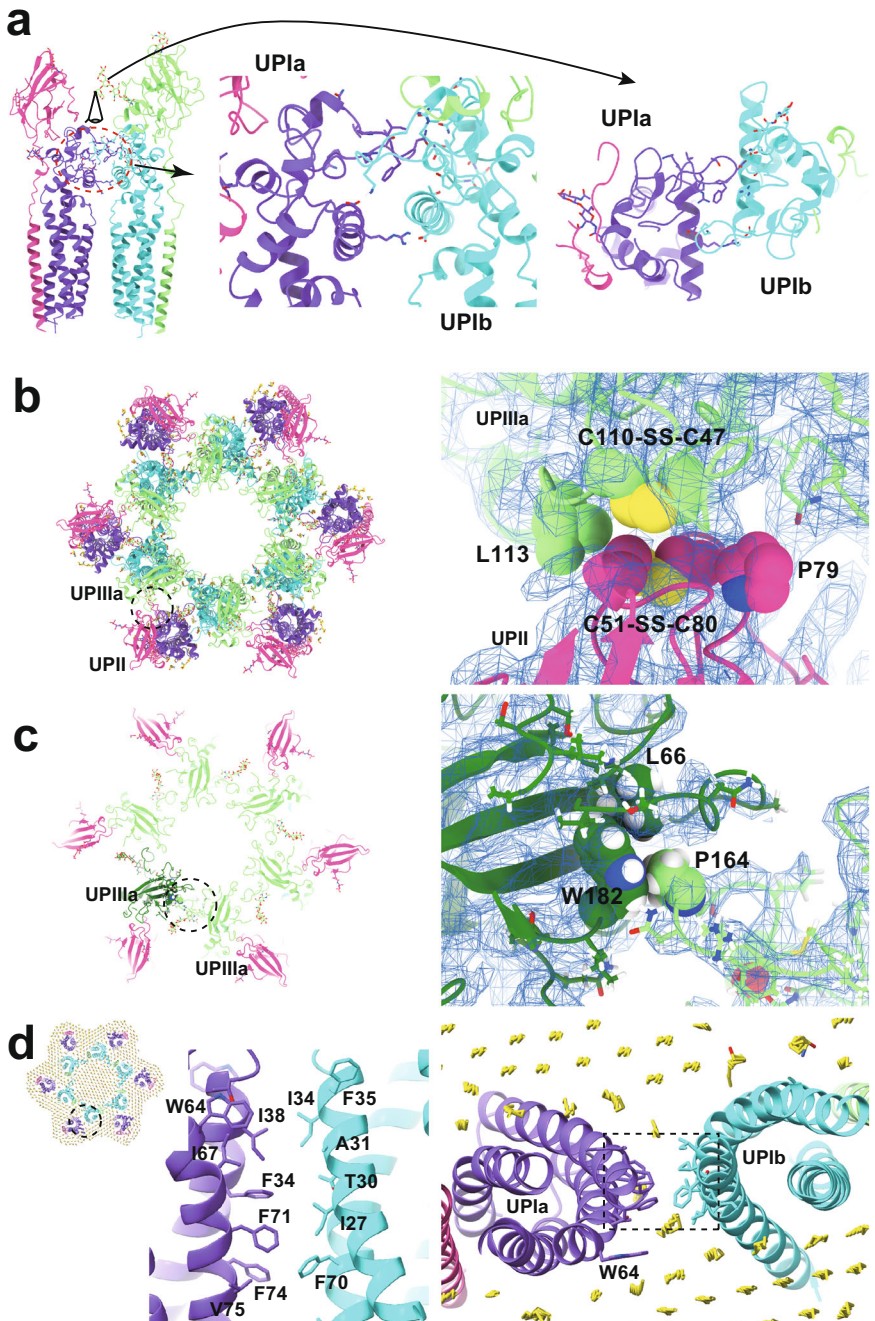

**Fig. 4 Inter-subunit interfaces. a** The interface between UPIa and Ib is characterized by two opposing helices (left, broken red circle). Side and top views are shown. **b** Hydrophobic interaction between UPII and IIIa. Disulfide bonds between Cys51-Cys80 of UPII and between Cys47-Cys110 of IIIa are facing each other, and the interface appears to be supported by Pro79 of UPII and Leu113 of IIIa. **c** IIIa-IIIa interface. Pro164 at the apex of the loop 137–161 of IIIa (arrows) inserts into the groove of Leu66 and Trp182 of the adjacent IIIa. **d** The interface between the transmembrane helices of Ia and Ib. At the pseudo-symmetric center of Ia and Ib (broken circle), a cluster of hydrophobic side chains mediates the Ia-Ib interaction.

the inner subdomain of the uroplakin hexameric complex based on the observation of additional densities in the difference map between the uroplakin structures with and without bound FimH, which interacts with glycosylated UPIa[19]. However, our reconstruction revealed that these "additional densities" are likely artifacts of the low-resolution EM maps, as the Ia/II heterodimer actually constitutes the outer subdomain of the uroplakin (Fig. 2).

Reports indicate that the binding of FimH to UPIa instigates conformational alterations within the transmembrane domains of uroplakin[16]. Contrarily, our analysis of the FimH-bound uroplakin structure did not reveal any notable conformational

modifications (Fig. 6b, d). The inherent flexibility of the carbohydrate chain coupled with the rapid on/off kinetics of FimH-mannose binding pose substantial challenges in visualizing the attached FimH and any resultant conformational changes[43–45]. The observed degradation in resolution within FimH-bound maps hints at potential alterations in the configuration of the uroplakin complex and/or lipid crystalline array. Variations in the fluidity or organization of the crystalline lipid membrane have been known to trigger signal transduction[46,47]. Given the apparent accommodation of FimH molecules within the interstitial space between two adjacent uroplakin complexes (Fig. S4d),

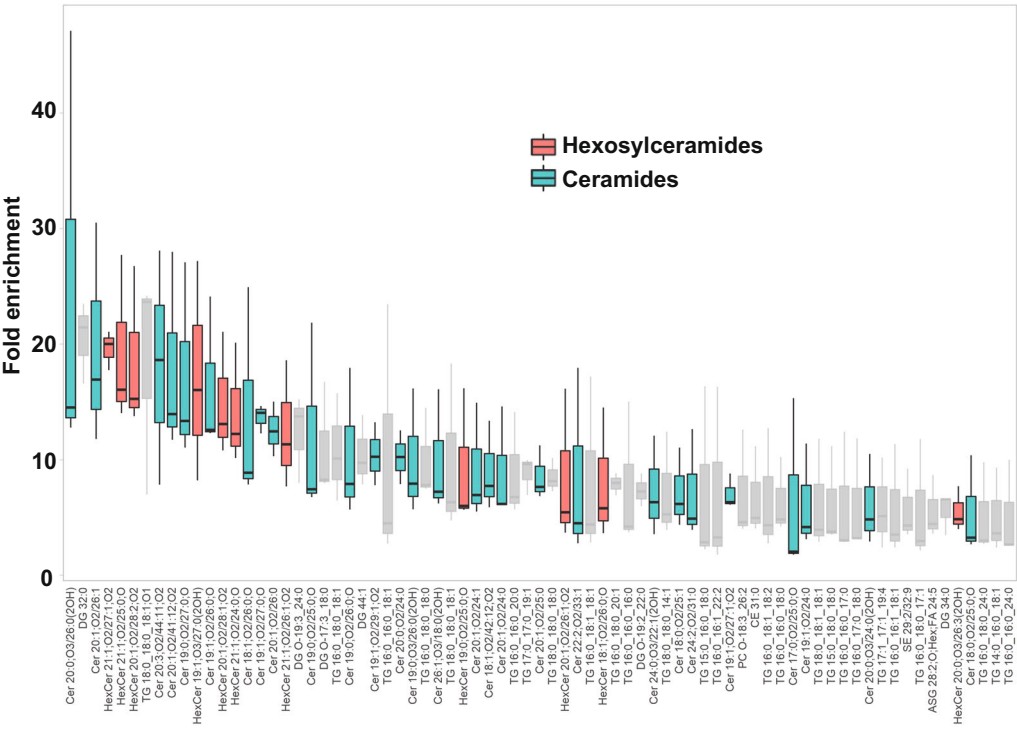

**Fig. 5 Lipidomic analysis of the AUM.** The chart displays the fold-enrichment of each lipid species detected in the sarkosyl-insoluble AUM fraction relative to the whole bladder scrape. Lipids were extracted from both sample types and analyzed using liquid chromatography-mass spectrometry (LC–MS). The peak heights were normalized by dividing them by the total signal for all identified lipids for each experiment. The fold-enrichment values indicate the difference in relative abundance of each lipid species in the sarkosyl-insoluble AUM fraction compared to the whole bladder scrape, with higher values suggesting an enrichment in the AUM fraction. The result for positive ion mode is shown. Data represent the mean fold-enrichment and standard deviation from three independent experiments ($n = 3$).

the ensuing molecular packing or crowding of FimH-bound uroplakin complexes could exert tension on the AUM, thereby eliciting cellular responses[48,49]. However, the precise mechanism through which the attachment of FimH to the flexible carbohydrate chain induces intracellular signaling remains an open question for further exploration[20].

Intriguingly, the prosequence of UPII persists within the mature uroplakin complex (Fig. S2c). The precursor form of UPII, which has a molecular weight of 29 kDa, is cleaved by the enzyme furin in the trans-Golgi network. This results in the separation of the N-terminal glycosylated prosequence and the "mature" UPII, which lacks glycosylation and has a molecular weight of 15 kDa[9,35,36]. However, it has been documented that the S2' fragment of SARS-CoV-2 spike protein remains associated with the molecule after cleavage by furin[50,51]. This observation holds true for the UPII prosequence. The question arises as to why only a single UPII band was observed in the electrophoresis of the AUM. One potential explanation is that the estimated molecular weight of the glycosylated UPII prosequence is roughly 14 kDa, and thus it may align with the 15 kDa band. Another possibility is that the heterogeneous nature of the glycosylation chains, which make up over half of the prosequence's mass, may not produce a distinguishable band in the electrophoresis.

The epithelial lining of the bladder demonstrates a distinct impermeability to water, protons, and urea, effectively preventing the infiltration of urinary substances into the surrounding tissue[52,53]. Depletion of uroplakin complexes from the urothelial apical membrane by UPIII knockout leads to a two-fold increase in water permeability. Yet, the urothelial barrier maintains a substantial degree of resistance to water and urea permeation[54]. These observations imply that the lipids and the uroplakin exert

complementary influences on the barrier properties of the urothelial apical membrane. Our lipidomic analysis revealed an enrichment of hexosylceramide in the AUM (Fig. 5)[41]. Previous studies have established that ceramides tend to associate with each other, and an accumulation of these molecules can form a liquid-crystalline phase[30,55]. The formation of ceramide-enriched crystalline lipid domains has also been linked to the clustering of CD95[56,57]. These findings suggest that the accumulation of hexosylceramides in the AUM may play a role in the formation of a hexagonal lattice structure of uroplakin[13].

Molecular dynamics simulations have predicted that sphingolipids form hexagonally arranged liquid-ordered phases[30,58]. The yeast plasma membrane $H^+$-ATPase, Pma1 hexamer, has been demonstrated to encircle a liquid-crystalline membrane microdomain[29]. Our findings represent the first direct visualization of hexagonally organized liquid-crystalline membrane domains in mammalian cells. Although the liquid-ordered phase of the lipid domain is believed to be essentially impermeable to water and ions, small molecules can traverse the membrane at the boundary between ordered and disordered domains[30,59]. It is noteworthy that the uroplakin-free hinge regions that connect the neighboring AUM plaques also exhibit resistance to sarkosyl and alkali treatments, indicating highly specialized structures[25]. These hinge regions may serve to prevent the penetration of small molecules at the boundary between the plaques.

While our team was successful in constructing a model of the uroplakin complex, we encountered a limitation in the resolution of the vertical dimension, which hindered our ability to confidently determine the precise positioning of the side chains. To overcome this challenge, we propose the utilization of high-resolution subtomogram averaging, as it has the capability to

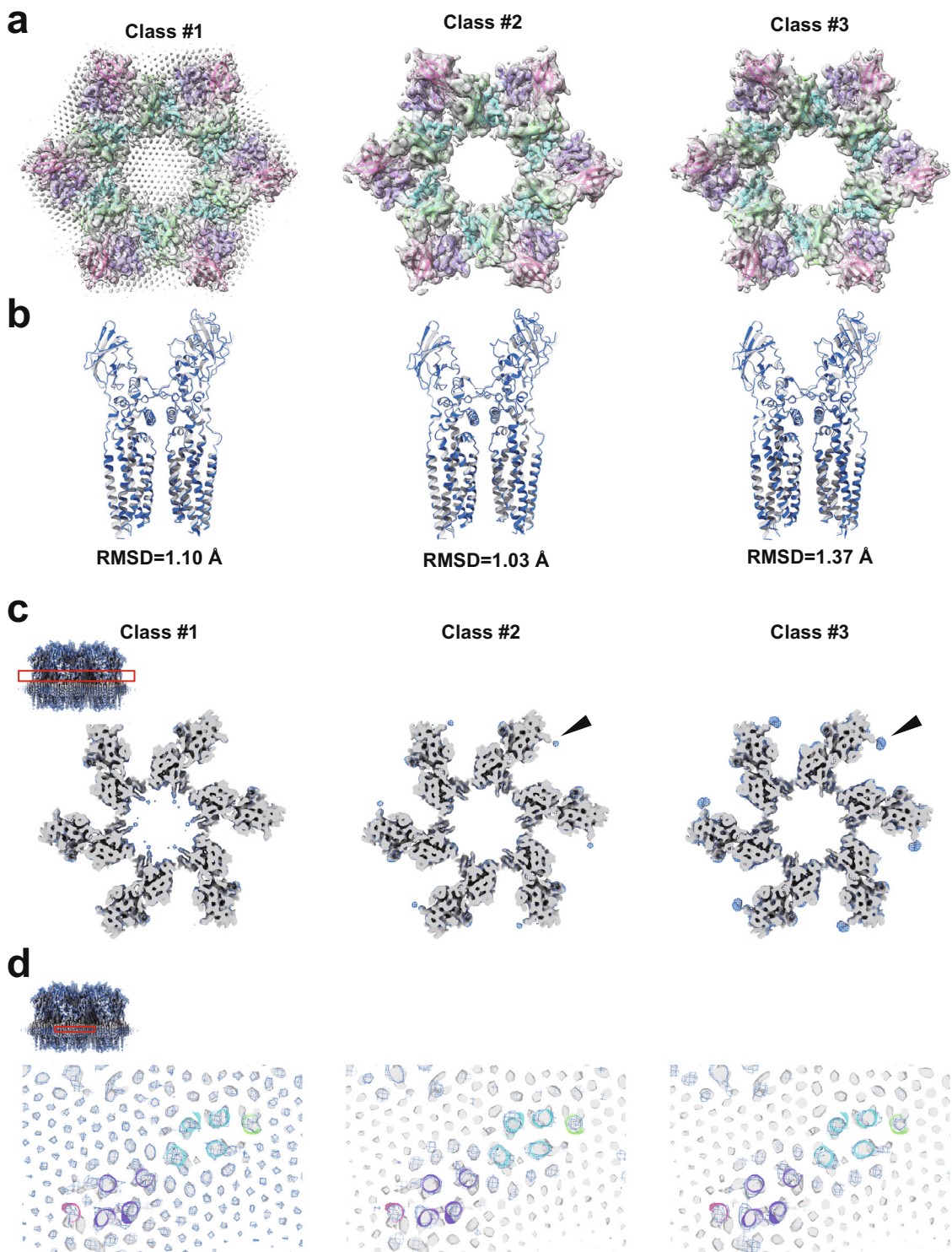

**Fig. 6 FimH-bound structure of the uroplakin complex. a** Map and fitted models of FimH-bound structures (classes #1-3). **b** Comparison between models of FimH-bound (blue) and unbound (gray). RMSDs between the two models were shown. **c**, **d** Slab sections of FimH-bound maps (classes #1-3) superimposed to FimH-unbound map. Blue mesh: FimH-bound; gray surface: FimH-unbound. Red rectangles of the insets indicate the positions of the slabs. **c** Slabs at the height of UPIa N169 show extra densities (arrowheads). **d** Slabs at the height of the transmembrane helices show no structural differences to the FimH-unbound structure, which is presented in atomic models.

bypass the overlap of molecules in the high-tilt views. However, to effectively implement this technique, it is imperative to devise an advanced algorithm that can accurately rectify the deformation of the AUM caused by irradiation.

## Methods

**Isolation of porcine AUM.** Fresh porcine bladders were obtained from a local slaughterhouse. The AUM was isolated according to the previous study with modifications[25]. The urothelium was

scraped from the luminal surface with a medicine scoop and suspended in ice-cold PBS. After centrifugation at $1500 \times g$ at 4 °C for 5 min, the pellets were homogenized with a Dounce glass homogenizer in buffer A (10 mM Hepes, pH 7.4, 1 mM EDTA, and protease inhibitor cocktail). After centrifugation at $2500 \times g$ at 4 °C for 10 min, the pellets were resuspended in buffer A, loaded onto a 1.6 M sucrose cushion, and centrifuged at $46,000 \times g$ at 4 °C for 30 min. The membrane fraction concentrated at the interface was collected, loaded onto a 1.6 M sucrose cushion, and centrifuged once again. The collected membrane fraction was resuspended in buffer A plus 2% sarkosyl and incubated at room temperature for 10 min. Sarkosyl-insoluble membranes were collected after centrifugation at $18,000 \times g$ at 4 °C for 30 min. The pellets were further washed by resuspension in 25 mM NaOH, and centrifuged at $18,000 \times g$ at 4 °C for 10 min. The pellets of the AUM were washed twice with buffer A and proceeded to electron microscopy.

**Purification of FimH-FimC complex**. The cDNA of *fimH* and its chaperone *fimC* genes were cloned from *E. coli* DH5a into pACYCDuet-1 bicistronic plasmid (Sigma-Aldrich, Burlington, MA), and the resulting co-expression plasmid was introduced to *E coli*. BL21 (DE3)[60]. The bacteria were grown at 37 °C in LB medium containing chloramphenicol (34 μg/ml). At an OD600 of 0.7, IPTG was added to a final concentration of 0.5 mM. The cells were further grown for 18 h at 15 °C, harvested by centrifugation, washed by PBS, and disrupted by sonication. After removing cell debris by centrifugation, the supernatant was applied to Ni-NTA resin (Qiagen, Germantown, MD) and eluted with 20 mM Tris–HCl pH 8.0, 0.3 M imidazole. Fractions containing FimH-FimC were pooled and loaded onto a gel filtration column (ProteinArk, Rotherham, UK) equilibrated with 20 mM Hepes-NaOH pH 7.4, 150 mM NaCl. Fractions containing FimH-FimC were dialyzed against buffer A and were concentrated using Vivaspin 2 (Sartorius, Göttingen, Germany). For cryo-EM, two times molar-excess FimH-FimC was mixed with the AUM (0.05 mg/ml) and incubated for 1 h at 4 °C before plunge-freezing.

**Cryo-electron microscopy of the AUM**. The AUM was suspended in buffer A at a concentration of 0.05 mg/ml. 3 μl of the sample was applied to freshly glow-discharged holey carbon grids, Quantifoil R1.2/1.3 Cu/Rh 200 mesh (Quantifoil Micro Tools GmbH, Großlöbichau, Germany), blotted from both sides for 3 s at 4 °C under 99% humidity and plunge frozen in liquid ethane using Vitrobot Mark IV (Thermo Fisher Scientific, Waltham, MA). Images were recorded using a Titan Krios G4 microscope at the University of Tokyo (Thermo Fisher Scientific) at 300 keV, a Gatan Quantum-LS Energy Filter (Gatan, Pleasanton, CA) with a slit width of 20 eV, and a Gatan K3 BioQuantum direct electron detector in the electron counting mode. The nominal magnification was set to $64,000 \times$ with a physical pixel size of 1.35 Å/pixel. Movies were acquired using the SerialEM software[61], and the target defocus was set to 2.5–4.5 μm for tomography and 1–3 μm for single particle analysis (SPA). For tomography, each movie was recorded for 0.18 s with a total dose of 1.26 electrons/Å² and subdivided into 10 frames. The angular range of the tilt series was from –60° to 60° with 3.0° increments using the dose-symmetric scheme[62] or the continuous scheme. The total dose for one tilt series acquisition is thus 50 electrons Å². For SPA, each movie was recorded for 6.7 s with a total dose of 50 electrons/Å² and subdivided into 50 frames. Specimens were tilted at 0, 30, 45, and 55°. The ratio of tilted images was 1:1:2:2 for 0, 30, 45, and 55°, respectively.

**Fluorescence microscopy of FimH-bound AUM**. The AUM suspension (1 mg/ml) was incubated with 0.1 mg/ml FITC-isothiocyanate (Dojindo Laboratories, Kumamoto, Japan) at room temperature for an hour, followed by four buffer A washes via centrifugation to obtain FITC-labeled AUM. Concurrently, FimH-FimC and BSA (2 mg/ml) were incubated with 0.2 mg/ml ATTO 590-NHS-ester (ATTO-TEC GmbH, Siegen, Germany) for an hour at room temperature. Unbound dye was removed using a PD-10 desalting column (cytiva, Marlborough, MA), and the proteins were concentrated using Vivaspin 2. The labeled AUM was then combined with FimH-FimC/BSA and incubated as described above.

Quantifoil R1.2/1.3 Cu/Rh 200 mesh grids were prepared by glow-discharging and blocking with 1 mg/ml BSA in buffer A for 3 min. A 3 μl sample was applied to each grid and plunge frozen as described above. Post-freezing, the grids were retrieved from the liquid ethane, thawed, and washed four times with buffer A. Finally, the grids were immersed in mounting buffer (buffer A supplemented with 50% glycerol and 0.1 M dithiothreitol) and visualized using a BX53 fluorescence microscope (Olympus/Evident, Tokyo, Japan).

**Data processing for tomography**. Movies were subjected to beam-induced motion correction, image alignment, CTF correction, and reconstruction by back-projection using the IMOD software package[63]. Tomograms were $8 \times$ binned and the centers of each uroplakin particle were manually selected using 3dmod tool. Volumes with 30-pixel³ dimensions were extracted from $8 \times$ binned tomograms and were averaged using the PEET software[64]. A randomly selected subtomogram was used for the initial reference. Alignments were repeated twice for $8 \times$ binned and once for $4 \times$ binned tomograms with 60-pixel³ dimensions. The averaged subtomogram was used for the reference in the subsequent SPA.

**Data processing for SPA**. Image analysis was executed utilizing CryoSPARC v4.2.1[65], as depicted in Fig S1a. Images, 4× binned at 128 pixels², were selected and extracted from motion-corrected and CTF-corrected micrographs, leveraging the projections of the mean subtomogram as templates. The particle stack underwent rigorous purification through five iterations of 2D classification and three cycles of heterogeneous refinement. Post local motion correction and CTF refinement, a homogenous refinement of the unbinned 512 pixels² particles produced a global resolution of 5 Å. Notably, at this juncture, the crystalline lipids remained undetected. To enhance the resolution, six uroplakin particles encircling the central particle were removed. The ensuing local refinement achieved a global resolution of 3.5 Å, revealing the crystalline lipids. The AUM's peripheries frequently exhibited folding, influencing the orientation distribution. Yet, the superimposed molecules affected the accuracy of the lateral view alignment. Resolutions in both transverse and vertical directions were ascertained using CryoSPARC's 3DFSC function (refer to Fig. S1c). The Root Mean Square Deviation (RMSD) between dual models was determined via UCSF ChimeraX. Regarding the FimH-bound structure, our analysis mirrored the methodology applied to the unbound structure, extending to particle subtraction and local refinement (Fig. S4c). To augment resolution, we adopted a reference-free and alignment-free 3D classification using CryoSPARC. Class #1 displayed the hexagonal crystalline lipid array, whereas Class #2 and #3 presented lipids proximate to the uroplakin complex. Class #3 distinctly exhibited additional densities adjacent to UPIa's Asn169. Efforts to amplify the resolution of these supplementary densities through targeted 3D classification did not result in any signal intensity enhancement.

**Table 1 Summary of data collection and model validation.**

| | Uroplakin complex (EMDB-36340) (PDB 8JJ5) |
|---|---|
| **Data collection and processing** | |
| Magnification | 64,000 |
| Voltage (kV) | 300 |
| Electron exposure (e–/Å$^2$) | 50 |
| Defocus range (μm) | 1.1–4.5 |
| Pixel size (Å) | 1.35 |
| Symmetry imposed | C6 |
| Initial particle images (no.) | 12,764,463 |
| Final particle images (no.) | 609,567 |
| Map resolution (Å) | 3.5 |
| FSC threshold | 0.143 |
| Map resolution range (Å) | 3.2–6.3 |
| **Refinement** | |
| Initial model used (PDB code) | AlphaFold prediction |
| Model resolution (Å) | 3.5 |
| FSC threshold | 0.143 |
| Model resolution range (Å) | 3.5–540.0 |
| Map sharpening B factor (Å$^2$) | −73.5 |
| **Model composition** | |
| Non-hydrogen atoms | 13,782 |
| Protein residues | 866 |
| Ligands | BMA:2, NAG:10, MAN:2 |
| **B factors (Å$^2$)** | |
| Protein | 59.32/134.09/93.14 |
| Ligand | 30.00/30.00/30.00 |
| **R.m.s. deviations** | |
| Bond lengths (Å) | 0.023 |
| Bond angles (°) | 2.466 |
| **Validation** | |
| MolProbity score | 1.74 |
| Clashscore | 2.55 |
| Poor rotamers (%) | 1.49 |
| **Ramachandran plot** | |
| Favored (%) | 89.25 |
| Allowed (%) | 10.75 |
| Disallowed (%) | 0.00 |

**Model building**. The initial model was generated using AlphaFold-multimer -multimer[28,66,67] and the *Sus scrofa* UPIa, UPIb, UPII, and UPIIIa sequences[68,69]. The AlphaFold-predicted model was refined using PHENIX (v.1.19.2-4158) real-space refinement tool with the secondary structure restraints and Ramachandran restraints on[70,71]. Carbohydrates were added to the Asn residues at the N-Glycosylation sites using Coot (v. 9.8.7)[72,73]. The models were further refined using UCSF Chimera, ChimeraX, and ISOLDE[74–77]. The refined model was validated using the comprehensive validation tool on PHENIX. Models of ceramide (CCD ID: 16 C) and sphingosine (CCD ID: SPH) were fitted manually and refined using ISOLDE. Maps and models are compiled in Supplemental Data 1. Parameters are summarized in Table 1.

**Lipidomic analysis**. The lipid composition of the isolated AUM was analyzed using LC–MS. The total urothelium membrane homogenate was used as a control. Lipids were extracted by adding 2-propanol (50:1 by volume to control homogenate or 500 μL to AUM preparation). After centrifugation (10,000 × g, 10 min, 4 C), the supernatant was further diluted (1:10) by 2-propanol, and 3 μL was analyzed by LC–MS. The LC–MS system consisted of an Acquity UPLC system (Waters, Milford, MA), connected to an Exactive mass spectrometer equipped with a HESI-II ion source (Thermo Scientific, Waltham, MA). An Acquity UPLC BEH C8 column (1.0 × 100 mm, Waters) was used for reversed-phase chromatography with binary gradient elution with mobile phase A (10 mM ammonium formate in acetonitrile/water/formic acid (60/40/0.1)) and B (10 mM ammonium formate in 2-propanol/acetonitrile/water/formic acid (90/9.5/0.5/0.1)) at a flowrate of 100 μL/min. Following time program was used [time (%B)]: 0 min (30%)–12 min (55%)–18 min (70%)–20 min (80%)–22 min (99%)–27 min (99%)–27.1 min (30%)–28.5 min (30%). MS was operated in following conditions to perform data-independent, all-ions fragmentation (AIF) MS/MS acquisition: ionization polarity, positive and negative; scan range, 100-2,000 m/z; mass resolution, 25,000; maximum injection time, 500 ms; AGC target, balanced ($1 \times 10^6$); in-source CID voltage, 0 V (off) and 35 V. MS-DIAL 4.9 software[78]. was used for peak detection, MS/MS deconvolution, alignment, integration, and lipid identification, of multiple samples. Lipid identification was based on searching lipid database provided with the software, with accurate mass tolerance of 0.005 Da and 0.01 Da for MS and MS/MS, respectively. For multiple candidates, those with highest score were adopted. No manual curation was performed for individual identifications, as the results were only used to estimate lipid classes abundant in AUM preparation. Peak intensities were normalized using built-in function of MS-DIAL, based on total signal for identified lipids (mTIC). R version 4.2 software was used for differential analysis using normalized peak intensity data exported from MS-DIAL. Fold-enrichment values were calculated as a ratio for AUM versus control using triplicate, independently prepared samples. See Supplemental Data 2 for numerical data used for generating boxplot in Fig. 5.

**Reporting summary**. Further information on research design is available in the Nature Portfolio Reporting Summary linked to this article.

## Data availability
The map, the model, and the raw movie data are available on the EMDB and EMPIAR under the following accession numbers: EMD-36340, PDB 8JJ5, and EMPIAR-11601. The source data for the box plot in Fig. 5 is given as Supplemental Data 2 and any remaining information can be obtained from the corresponding author upon reasonable request.

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

## Acknowledgements

We thank Mrs. Natsuko Maruyama (University of Yamanashi) for technical assistance. This research is partially supported by Platform Project for Supporting Drug Discovery and Life Science Research (Basis for Supporting Innovative Drug Discovery and Life Science Research (BINDS)) from Japan Agency for Medical Research and Development (AMED) under Grant Numbers JP22ama121002 and JP23ama121002 (support number 2639). This work was supported by the Takeda Science Foundation (to T.O.), the Daiichi Sankyo Foundation of Life Science (to T.O.), the Japan Society for the Promotion of Science (KAKENHI Grant numbers JP21H02654, JP22H05538 (T.O.), JP21H05248 (M.K.)), and the Naito Foundation (to T.O.). Molecular graphics and analyses performed with UCSF ChimeraX, developed by the Resource for Biocomputing, Visualization, and Informatics at the University of California, San Francisco, with support from National Institutes of Health R01-GM129325 and the Office of Cyber Infrastructure and Computational Biology, National Institute of Allergy and Infectious Diseases.

## Author contributions

T.O., H.Y., and M.K. designed the research; T.O., Y.K., and H.Y. performed experiments and analyzed data. T.O., Y.K., and M.K. wrote manuscript.

## Competing interests

The authors declare no competing interests.

## Ethical approval

This research strictly followed the ethical guidelines and standards established by our institutions, University of Yamanashi and the University of Tokyo, located in Japan. The entire research process, from study design and implementation to data ownership and the authorship of publications, was carried out exclusively by the authors listed, all of whom are researchers at the aforementioned institutions. As this research was conducted in Japan, by Japanese researchers, it is both locally relevant and compliant with local standards and regulations. The roles and responsibilities of each researcher were established and agreed upon at the outset of the research process. We did not undertake capacity-building plans for external researchers, as our team had all the necessary capabilities to conduct this research. The research has not been restricted or prohibited within our local context. Therefore, there was no need for any special exceptions or permissions to be sought from local stakeholders. Our research does not involve any elements that could lead to stigmatization, incrimination, discrimination, or personal risk to participants. It also does not pose any health, safety, security, or other risk to the researchers involved. The research did not involve the transfer of any biological materials, cultural artefacts, or associated traditional knowledge out of Japan. Hence, no benefit sharing measures were discussed or required. In the formulation of our research and the analysis of our results, we have considered and cited local and regional research relevant to our study.
