## [Peer Review File · Communications Biology]

Reviewers' comments:

Reviewer #1 (Remarks to the Author):

This study Yanagisawa et al. presents the cryoEM structure of the uroplakin complex, which corrects to previously misinterpreted low-resolution map. The authors also determined a structure of the uroplakin complex binding with FimH/FimC, and overthrow the previous dominant model that FimH binding triggers TM conformational changes. However, my concerns are whether the cryoEM map is not strong enough to support the model proposed.

Main concerns

1. The authors claimed a 3.5 Å overall resolution, however, the visual inspection of the map makes me doubt this claim, as most sidechains are not visible. Cryosparc intends to claim a higher resolution, especially for preferred orientation structures. I would not trust this 3.5 Å resolution claim judging by the quality of the map.

2. A C6 symmetry was applied at an early stage of data processing. Are the crystalline lipid density artifacts introduced by C6 symmetry? Have the authors processed data without symmetry? Can you still observe crystalline lipid density without C6 symmetry? And why only lipids from the outer leaflet are visible in the map?

3. In Supplementary fig 3, the authors present the difference map of uroplakin along compared with the binding of FimH-FimC. Unfortunately, I cannot tell which densities are contributed by FimH/FimC. Can authors highlight the density of FimH/FimC? If the authors cannot see the density of FimH/FimC, you cannot simply assume that the FimH/FimC are present in your map because you mix uroplakin and Fim in your samples. In my laboratory, we have met many cases that complexes dissociate after freezing grids.

For other small concerns, I would suggest using white color instead of black color as the background. Overall, I'm not enthusiastic to publish this manuscript with the quality of the cryoEM map.

Reviewer #2 (Remarks to the Author):

Summary of Findings:

In this study, the authors utilized cryo-electron microscopy (cryo-EM) to investigate the molecular architecture of the uroplakin complex within the asymmetric unit membrane (AUM) of the urothelium, a distinct epithelial tissue lining the urinary tract. The authors achieved the highest resolution structures of the uroplakin complex reported to date (3.5 Å), revealing its location within hexagonally arranged crystalline lipid membrane domains rich in hexosylceramides. Based on their cryo-EM structures, the authors correct previously reported subunit arrangements within the uroplakin complex, identify the crucial *Escherichia coli* binding site involved in urinary tract infections, and highlight the role of hexosylceramides in the AUM's barrier function.

Overall Impression:

This study fills an important knowledge gap by investigating the molecular architecture of the uroplakin complex and the role of liquid-crystalline lipids within the urothelial membrane. The biggest contribution of this study is visualization of the uroplakin complex in its native membrane in high-resolution and correction of a previously reported model of subunit arrangement that was built using a lower-resolution EM data. In addition, the discovery of hexagonally arranged crystalline lipid domains rich in hexosylceramides within the AUM, along with the identification of the *E. coli* binding site on the uroplakin complex, carries significant biological implications, providing insights into the urothelium's permeability barrier function and its susceptibility to urinary tract infections. While the manuscript is

well-written and the authors' findings make significant contribution to the field by clarifying the basic molecular architecture of the uroplakin complex, certain claims made in the manuscript, particularly regarding the cryo-EM structure of the uroplakin bound to the E. coli protein FimH, are not well supported. Also, the authors are strongly encouraged to consider rearranging and revising some of their figures to present their findings more clearly and avoid potential confusion.

Specific Comments:

1. The authors claim that binding of the E. coli protein FimH does not induce structural change in the uroplakin complex, based on the comparison of the cryo-EM structures they determined for uroplakin with and without FimH. However, the cryo-EM map of FimH-bound uroplakin is not provided and the data presented in the manuscript is insufficient and not very convincing. (A) The superimposition of the two cryo-EM maps in figure S3A is unclear and very hard to see. While a small extra density is visible, it is very small and it certainly can't be ruled out that it is just a noise in the map. (B) Considering that FimH was added to the uroplakin complex prior to freezing the grids without confirming formation of the complex, it is possible that FimH did not bind to uroplakin at all or the structure represents a mixture of uroplakin bound and unbound to FimH.

2. The authors claim that the cryo-EM structure of FimH-bound uroplakin "...indicates an alteration in the signal intensities of crystalline lipids near the Ia/II heterodimer". They refer to the difference map in figure S3A to support this claim, but this does not seem to be a difference map (one map subtracted from another), but rather a superimposition of a cryo-EM map and a model (not mentioned which is for which sample). Either way, the difference between the two are very small to be significant (at least from the way it is presented in the figure).

3. All figures related to FimH-uroplakin complex is in the supplementary data even though that is one of the main findings according to the authors.

4. The authors should provide a cryo-EM workflow for FimH-uroplakin complex structure as they did for the structure of uroplakin alone.

5. All the figures showing structural features of uroplakin complex have dark blue cryo-EM maps shown against a black background. This makes it very hard for readers to see and evaluate how well the authors' models fit into the cryo-EM maps (and thus support the authors' interpretation of their structural data). Using a white background in all the figures is highly recommended.

6. Figure 4 seems to be more appropriate for the supplementary data.

Reviewer #3 (Remarks to the Author):

In this paper, Yanagisawa et al., studied the 3D structure of uroplakin complex using cryoEM. Their findings offered valuable insights into the molecular architecture of urothelial membrane and corrected a misconception derived from an earlier model. Although the quality of the reconstructed map is not great along the direction perpendicular to the membrane due to some technical limitations, the results are the best obtained so far. I will support the publication if the authors make further efforts to clarify the following points:

1. It is recommended to show the map and model of some individual transmembrane helices, which will help the readers to assess the quality of reconstruction.
2. 3D FSC curves should be shown. They are mentioned in the methods but not shown in the figures.

3. Are the crystalline lipids visible without applying C6 symmetry? The authors modeled mostly single-chain sphingolipids instead of ceramides, which leads to the question that whether enforcing the symmetry would cause some artifacts. This reviewer is convinced that crystalline lipids play an essential role in the structure of uroplakin, but not sure how to account for the ratio of sphingolipids versus ceramides in the model.
4. The riding hydrogens are helpful for structure refinement, but it is recommended to hide them for structure presentation.
5. How does the final model differ from the complex structure predicted by AlphaFold multimer?
6. It is surprising that the FimH+ structure does not show any significant additional density. Are there any other evidence showing that FimH-FimC are actually present? Could the potential complex dissociate upon plunge-freeze?

Responses to the reviewers (Reviewers' comments and **answers by the authors**)

Reviewer #1:

Main concerns

1. The authors claimed a 3.5 Å overall resolution, however, the visual inspection of the map makes me doubt this claim, as most sidechains are not visible. Cryosparc intends to claim a higher resolution, especially for preferred orientation structures. I would not trust this 3.5 Å resolution claim judging by the quality of the map.

Answer:

In response to your concerns regarding the claimed resolution of our cryo-electron microscopy data, we appreciate your careful scrutiny and understand your reservations. We have revised our description of the resolution in the Abstract to more accurately reflect the data. The revised statement reads as follows:

"In this study, we utilized cryo-electron microscopy to elucidate the three-dimensional structure of the uroplakin complex within the porcine AUM. While the global resolution achieved was 3.5 Å, we acknowledge that due to orientation bias, the resolution in the vertical direction was determined to be 6.3 Å."

This revised statement provides a more nuanced description of our data, acknowledging the limitations imposed by orientation bias. We trust that this clarification addresses your concerns and provides a more accurate representation of our findings.

2. A C6 symmetry was applied at an early stage of data processing. Are the crystalline lipid density artifacts introduced by C6 symmetry? Have the authors processed data without symmetry? Can you still observe crystalline lipid density without C6 symmetry? And why only lipids from the outer leaflet are visible in the map?

Answer:

In response to the query regarding the application of C6 symmetry and its potential impact on the visualization of crystalline lipids, we have indeed processed the data without the imposition of C6 symmetry. The resultant visualization of the crystalline lipids is presented in Fig. S1A and Supplemental Data1 (C1map.mrc).

As for the observation of lipids exclusively from the outer leaflet, this can be attributed to the asymmetric distribution of cholesterol within the raft domain. Specifically, the concentration of cholesterol in the inner leaflet is approximately 12 times less than that in the outer leaflet, as reported by Liu et al. (Nat. Chem. Biol., 2017). Consequently, it is plausible that the lipids in the

inner leaflet exhibit a higher degree of disorder compared to those in the outer leaflet. This observation aligns with previous research on yeast ATPase Pma1 conducted by Zhao et al. (Nat. Comm., 2021), where crystalline lipids were visualized solely in the outer leaflet. Therefore, the visibility of lipids from the outer leaflet in our study is consistent with established literature and can be explained by the differential cholesterol concentration across the leaflets.

3. In Supplemental fig 3, the authors present the difference map of uroplakin along compared with the binding of FimH-FimC. Unfortunately, I cannot tell which densities are contributed by FimH/FimC. Can authors highlight the density of FimH/FimC? If the authors cannot see the density of FimH/FimC, you cannot simply assume that the FimH/FimC are present in your map because you mix uroplakin and Fim in your samples. In my laboratory, we have met many cases that complexes dissociate after freezing grids.

Answer:

In response to the query regarding the discernibility of FimH/FimC densities, we appreciate the concern raised. We have employed a rigorous experimental approach to confirm the presence of FimH/FimC on the AUM.

Firstly, we utilized SDS-PAGE to confirm the attachment of FimH/FimC to the AUM. As shown in Fig. S4A, the presence of bands corresponding to FimH and FimC (indicated by black arrowheads) post-centrifugation confirms their association with the AUM.

Secondly, we employed fluorescence microscopy to further validate this association. The AUM was labeled with FITC-isothiocyanate and FimH-FimC with ATTO 590-NHS-ester. Upon mixing and incubation, the samples were applied to Quantifoil grids and plunge-frozen. Post-thawing, the grids were visualized using a fluorescence microscope. As demonstrated in Fig. S4B, the co-localization of FITC-labeled AUM and ATTO 590-labeled FimH-FimC signals confirms the presence of FimH/FimC on the AUM. In contrast, ATTO 590-labeled BSA did not attach to the AUM, further validating the specificity of the FimH-FimC and AUM interaction.

Thus, while we acknowledge the potential for complex dissociation post-freezing, our combined use of SDS-PAGE and fluorescence microscopy provides robust evidence for the presence of FimH/FimC on the AUM in our samples.

For other small concerns, I would suggest using white color instead of black color as the background.

Answer:

We have taken your suggestion into account and adjusted the background to white for all figures in the manuscript. We believe this change enhances the clarity and readability of our data presentation.

Reviewer #2:

Specific Comments:

1. The authors claim that binding of the *E. coli* protein FimH does not induce structural change in the uroplakin complex, based on the comparison of the cryo-EM structures they determined for uroplakin with and without FimH. However, the cryo-EM map of FimH-bound uroplakin is not provided and the data presented in the manuscript is insufficient and not very convincing. (A) The superimposition of the two cryo-EM maps in figure S3A is unclear and very hard to see. While a small extra density is visible, it is very small and it certainly can't be ruled out that it is just a noise in the map.

Answer:

Thank you for your comments regarding the evidence for the lack of structural changes in the uroplakin complex upon FimH binding. We have taken steps to provide a clearer presentation of our data and methodology.

Firstly, we would like to clarify that we performed a 3D classification using CryoSPARC for the FimH-bound structure. The cryo-EM maps resulting from this classification are included in the supplemental data (FimHplus_Class1, 2, 3.mrc). We acknowledge that the superimposition of the two cryo-EM maps in Figure S3A was not as clear as intended. However, from the 3D classification, particularly in classes #3, we observed additional densities adjacent to UPIa N169 (Figure 7B, arrowheads). These densities are more pronounced and distinct than what might be attributed to mere noise in the map.

It's crucial to understand that the interaction between FimH and the uroplakin complex is unique. FimH binds to the mannose moiety at the tip of a highly flexible carbohydrate chain. In our structure, we visualized only the very root of this carbohydrate chain at Asn169 of UPIa. Given this interaction's nature, it's plausible that FimH doesn't form a rigid contact with the uroplakin complex, which might explain the absence of significant structural changes in the uroplakin complex upon FimH binding. Moreover, our findings suggest that FimH binding may subtly modify the uroplakin-lipid array arrangement without inducing major changes in the uroplakin complex conformation. This observation is consistent with the FimH-bound map displaying a lower resolution and weaker lipid densities compared to the FimH-unbound map (Figure S4C, pre-3D classification). However, post our 3D classification, we obtained a higher resolution map (Fig. 7A and S4C, class #1) that showed no discernible structural differences between the FimH-bound and unbound maps (Figure 7A and B).

In summary, our revised data presentation, combined with the 3D classification results, offers compelling evidence supporting our claim that FimH binding does not induce significant structural changes in the uroplakin complex.

(B) Considering that FimH was added to the uroplakin complex prior to freezing the grids without confirming formation of the complex, it is possible that FimH did not bind to uroplakin at all or the structure represents a mixture of uroplakin bound and unbound to FimH.

Answer:

In response to your concerns about the potential detachment of FimH from the uroplakin complex during the freezing process, we have taken several measures to ensure the validity of our findings. Firstly, we have used SDS-PAGE to confirm the binding of FimH to the AUM. As shown in Fig. S4A, the presence of bands corresponding to FimH post-centrifugation provides strong evidence of its association with the AUM. The absence of these bands would have indicated their dissociation during the process.

Secondly, to further validate this association, we employed fluorescence microscopy. The AUM was labeled with FITC-isothiocyanate and FimH with ATTO 590-NHS-ester. After incubation, the samples were applied to Quantifoil grids and plunge-frozen. The grids were then visualized post-thawing, and as demonstrated in Fig. S4B, the co-localization of FITC-labeled AUM and ATTO 590-labeled FimH signals confirms the presence of FimH on the AUM.

We acknowledge the possibility of complex dissociation post-freezing, a phenomenon that is not uncommon in cryo-EM studies. However, the combined use of SDS-PAGE and fluorescence microscopy in our study provides robust evidence for the presence of FimH on the AUM in our samples. Furthermore, the absence of ATTO 590-labeled BSA attachment to the AUM underscores the specificity of the FimH-AUM interaction.

In conclusion, while we appreciate the potential for variability in complex formation, our multifaceted experimental approach provides compelling evidence for the presence of FimH on the AUM, thereby supporting the validity of our findings.

2. The authors claim that the cryo-EM structure of FimH-bound uroplakin “..indicates an alteration in the signal intensities of crystalline lipids near the Ia/II heterodimer”. They refer to the difference map in figure S3A to support this claim, but this does not seem to be a difference map (one map subtracted from another), but rather a superimposition of a cryo-EM map and a model (not mentioned which is for which sample). Either way, the difference between the two are very small to be significant (at least from the way it is presented in the figure).

Answer:

In response to your comments regarding the interpretation of the difference map in Figure S3A, we appreciate your feedback and have made adjustments accordingly. We acknowledge that the differences between the two maps were indeed subtle and may not have been readily discernible. To address this, we have revised Figure 7 (previously supplemental figure 3) to provide a clearer comparison. The figure now includes slab sections of FimH-bound maps (classes #1-3) superimposed onto the FimH-unbound map. In these revised figures, the blue mesh represents the FimH-bound structure, while the gray surface represents the FimH-unbound structure. At the height of UPIa N169, we identified additional densities (indicated by arrowheads) in the FimH-bound maps (Figure 7B, top). These additional densities are hypothesized to represent bound FimH components and/or FimH-stabilized carbohydrate chains.

We agree that the differences between the FimH-bound and unbound maps are minimal. However, our findings suggest that FimH binding may subtly modify the uroplakin-lipid array arrangement without inducing significant changes in the uroplakin complex conformation. This observation is consistent with our initial claim and is now better represented in the revised Figure 7.

We hope that these revisions and clarifications adequately address your concerns and provide a more convincing presentation of our findings.

3. All figures related to FimH-uroplakin complex is in the supplemental data even though that is one of the main findings according to the authors.

Answer:

We acknowledge your point regarding the significance of the FimH-uroplakin complex in our findings. In light of your feedback, we have relocated the relevant figures from the supplemental data to the main body of the manuscript (Figure 7).

4. The authors should provide a cryo-EM workflow for FimH-uroplakin complex structure as they did for the structure of uroplakin alone.

Answer:

Accordingly, we have added the workflow for the FimH-bound structure to Supplemental Figure 4C.

5. All the figures showing structural features of uroplakin complex have dark blue cryo-EM maps

shown against a black background. This makes it very hard for readers to see and evaluate how well the authors' models fit into the cryo-EM maps (and thus support the authors' interpretation of their structural data). Using a white background in all the figures is highly recommended.

Answer:

We understand your concern regarding the visibility of our figures due to the color contrast between the cryo-EM maps and the background. To enhance readability and clarity, we have changed the background color to white in all figures. We believe this adjustment will make it easier for readers to evaluate the fit of our models into the cryo-EM maps.

6. Figure 4 seems to be more appropriate for the supplemental data.

Answer:

We appreciate your suggestion regarding the placement of Figure 4. In response, we have relocated Figure 4 to the supplemental data, where it is now presented as Supplemental Figure 3.

Reviewer #3:

1. It is recommended to show the map and model of some individual transmembrane helices, which will help the readers to assess the quality of reconstruction.

Answer:

Accordingly, we have added the densities and models of the transmembrane helices to Supplemental Figure 1D.

2. 3D FSC curves should be shown. They are mentioned in the methods but not shown in the figures.

Answer:

We have added the 3D FSC plot to Supplemental Figure 1B (right).

3. Are the crystalline lipids visible without applying C6 symmetry? The authors modeled mostly single-chain sphingolipids instead of ceramides, which leads to the question that whether enforcing the symmetry would cause some artifacts. This reviewer is convinced that crystalline lipids play an essential role in the structure of uroplakin, but not sure how to account for the ratio of sphingolipids versus ceramides in the model.

Answer:

In response to your concerns regarding the visibility of crystalline lipids without enforcing C6 symmetry and the modeling of single-chain sphingolipids versus ceramides, we appreciate your insightful observations.

We have indeed processed our data without applying C6 symmetry. The resultant visualization of crystalline lipids, which is presented in Fig. S1A (Supplemental Data1, C1map.mrc), confirms their presence without the imposition of symmetry constraints. This approach ensures that our observations are not artifacts of enforced symmetry.

As for the modeling of single-chain sphingolipids versus ceramides, we acknowledge the complexity of lipid composition in biological membranes. The ratio of sphingolipids to ceramides in our model is a representation and may not reflect the exact in vivo ratio. We have noted this in the figure legends to clarify that the models of sphingolipids and ceramides are placed tentatively and that the precise identities of these lipids within the structure remain uncertain.

We appreciate your understanding of the limitations inherent in such modeling and your recognition of the essential role of crystalline lipids in the structure of uroplakin. We believe our findings provide valuable insights into the structure and function of this complex, while also highlighting areas for further investigation.

4. The riding hydrogens are helpful for structure refinement, but it is recommended to hide them for structure presentation.

Answer:

We agree that hiding riding hydrogens can enhance the clarity of our structure presentation. We have made this adjustment and the revised figures now exclude the display of hydrogens.

5. How does the final model differ from the complex structure predicted by AlphaFold multimer?

Answer:

We have included the AlphaFold predicted model in Supplemental Data 1 (Alphafold_prediction.pdb). The root-mean-square deviation (RMSD) between the predicted model and our final model is 2.52 Å.

6. It is surprising that the FimH+ structure does not show any significant additional density. Are there any other evidence showing that FimH-FimC are actually present? Could the potential complex dissociate upon plunge-freeze?

Answer:

In response to your concerns regarding the discernibility of FimH/FimC densities and the possibility of complex dissociation upon plunge-freezing, we appreciate your insightful comments.

We have indeed taken rigorous steps to confirm the presence of FimH/FimC on the AUM. Firstly, we utilized SDS-PAGE to verify the attachment of FimH/FimC to the AUM. As shown in Fig. S4A, the presence of bands corresponding to FimH and FimC (indicated by black arrowheads) post-centrifugation confirms their association with the AUM.

Secondly, we employed fluorescence microscopy to further validate this association. The AUM was labeled with FITC-isothiocyanate and FimH-FimC with ATTO 590-NHS-ester. Upon mixing and incubation, the samples were applied to Quantifoil grids and plunge-frozen. Post-thawing, the grids were visualized using a fluorescence microscope. As demonstrated in Fig. S4B, the co-localization of FITC-labeled AUM and ATTO 590-labeled FimH-FimC signals confirms the presence of FimH/FimC on the AUM. In contrast, ATTO 590-labeled BSA did not attach to the AUM, further validating the specificity of the FimH-FimC and AUM interaction.

While we acknowledge the potential for complex dissociation post-freezing, our combined use of SDS-PAGE and fluorescence microscopy provides robust evidence for the presence of FimH/FimC on the AUM in our samples. This evidence, we believe, convincingly addresses your concerns and underscores the validity of our findings.

Reviewers' comments:

Reviewer #1 (Remarks to the Author):

The authors have addressed several concerns raised in the initial review regarding the FimH-bound uroplakin structure. The reviewer raised doubts about the formation of the FimH-uroplakin complex, lack of clear cryo-EM map for FimH-bound uroplakin, and about the proper interpretation of the structural data presented.

In response, the authors have provided additional data, revised figures, and explanations to clarify their methodology and findings. Most notably, they addressed the concerns about the complex formation by using SDS-PAGE and fluorescence microscopy to confirm the presence of FimH on the uroplakin complex on a cryo-EM grid. The revised figures also make it easier to locate where the additional density for FimH is found in the cryo-EM map of the complex. Overall, the additional data and figures help present their findings more convincingly.

Reviewer #2 (Remarks to the Author):

The authors have made appreciable efforts to address my concerns raised during the initial review. It is recommended that the manuscript be published without further review.

Here are two minor suggestions:

1. The labels of supplementary Figure 4B are a bit confusing (two panels with "AUM FITC").
2. The authors should consider depositing the raw data to a public database such as EMPAIR so that other researchers especially method developers may further explore new approaches for better results.

Reviewer #3 (Remarks to the Author):

The uroplakin complexes are vital in ensuring urinary tract integrity and guarding against infections. Yet, high-resolution structural insights have been scarce. Haruaki et al. employed cryo-electron microscopy to disclose the structure of the uroplakin complex within the porcine AUM. Their work underscores the uroplakin complex's arrangement within a crystalline lipid milieu, abundant in hexosylceramides. They also rectified a past oversight, pinpointing a bacterial binding domain previously missed. These insights deepen our understanding of urothelium's functions. However, to get published, the paper needs to be largely rewritten, the conclusion needs to be drawn with care, and figures will need to be generated with better clarity. Moreover, there are discrepancies between the reported and actual resolutions of the cryo-EM maps; Thus, the data's resolution appears insufficient, and interpretations seem overstated.

Key Issues:

1. The binding of FimH is indicated through methods like SDS-page and fluorescence microscopy. However, these only provide a pre-freezing state and aren't directly related to EM sample freezing. Hence, any binding that's weak or non-specific may separate during freezing. Clear density with distinct shapes is essential to confirm direct binding, without that the comparison results between the presence and absence of FimH don't seem valuable. Consequently, the study's point that uroplakin's structure remains unaltered upon FimH binding isn't compelling.
2. The resolution derived from the FSC curve in Supplementary Figure 1B appears closer to 7 angstroms without masking (FSC=0.5), which deviates too much from the declared resolution though

gold stand is applied.

3. Similar to last issue, in Supplementary Figure 4, if we use the FSC curve without masking, the resolution seems around 10 angstroms at FSC 0.5, differing significantly from the reported 3.7 angstroms. This is concerning since the paper's primary data doesn't match the claimed high resolution. This could lead to overinterpretation of the data.

4. Processing data with C6 symmetry followed by refinement without symmetry could pose challenge for accurate particle alignment, making the resultant C1 map, especially at low resolution, hard for accurate lipid density interpretation. (Admittedly, those stretches might be lipids). Preferred orientation plus C6 symmetry applied at early stage may also lead to such phenomenon if only several lipid density present; thus the claim for crystalline lipid does not seem solid, I would suggest move it to the discussion session.

5. Again, Supplementary Figure 2 suggests that the map's resolution isn't sufficient for modelling side chains. Supplementary Figure 3 has similar issues.

6. As the current resolution do not allow accurate assignment of side chains, all Figures with detailed interactions need to be moved.

7. In Fig 4D, the separation between D150 of UPIa and K197 of UPIb is a mere 1.8 angstrom. Could this be indicative of a covalent bond? Similarly, the interaction between N195 and Q130 is also measured at just 1.8 angstrom.

Reviewers' comments and **authors' replies**:

Reviewer #1 (Remarks to the Author):

No specific suggestions by this reviewer.

Reviewer #2 (Remarks to the Author):

Here are two minor suggestions:

1. The labels of supplementary Figure 4B are a bit confusing (two panels with “AUM FITC”).

Answer: Thank you for pointing out the confusion in the labels of Supplementary Figure 4B. We appreciate your attention to detail, and we have revised the figure to address this issue. The captions have been updated to include "AUM+FimH" and "AUM+BSA," which should clarify the distinction between the panels.

2. The authors should consider depositing the raw data to a public database such as EMPAIR so that other researchers especially method developers may further explore new approaches for better results.

Answer: We wholeheartedly agree with your suggestion to deposit the raw data to a public database, and we have taken action to do so. The raw movie files have been deposited to EMPIAR, and they are currently under process. The raw data will be available under the entry ID of EMPIAR-11601.

Reviewer #3 (Remarks to the Author):

Key Issues:

1. The binding of FimH is indicated through methods like SDS-page and fluorescence microscopy. However, these only provide a pre-freezing state and aren't directly related to EM sample freezing. Hence, any binding that's weak or non-specific may separate during freezing. Clear density with distinct shapes is essential to confirm direct binding, without that the comparison results between the presence and absence of FimH don't seem valuable. Consequently, the study's point that uroplakin's structure remains unaltered upon FimH binding isn't compelling.

Answer: Thank you for your insightful comments. We understand your concern regarding the evidence for FimH binding to the AUM in our cryo-EM studies.

We would like to clarify that our fluorescence microscopy was conducted on grids that **were frozen and then thawed**, simulating the conditions of cryo-EM sample freezing. This freeze-thaw approach was specifically designed to demonstrate that FimH binding to the AUM is maintained under cryo-EM conditions.

We believe this methodology supports our conclusion that the uroplakin structure remains unaltered upon FimH binding. We hope this clarification addresses your concern, and we appreciate your continued engagement with our work.

2. The resolution derived from the FSC curve in Supplementary Figure 1B appears closer to 7 angstroms without masking (FSC=0.5), which deviates too much from the declared resolution though gold stand is applied.

Answer: Thank you for your careful examination of our manuscript and for raising important concerns about the resolution of our maps. We recognize the significance of your observations, and we sincerely apologize if our previous version did not clearly convey the methodology we used for resolution estimation.

In our study, we utilized a very large box size (512 pixel³) that includes the central uroplakin complex of interest along with surrounding and partial complexes. This large box was essential for particle subtraction to achieve better alignment of the central complex. We then used a smaller mask for resolution estimation, focusing specifically on the central complex of interest. The revised Supplementary Figure 1A illustrates the 512-pixel³ box and the mask used, providing a visual context for our approach.

We understand that our initial presentation may have led to confusion, and we are committed to revising our manuscript to ensure that our approach is transparent and understandable.

Your expertise and critical insights are invaluable to us, and we are more than willing to engage in a constructive dialogue or make necessary adjustments to address your concerns.

3. Similar to last issue, in Supplementary Figure 4, if we use the FSC curve without masking, the resolution seems around 10 angstroms at FSC 0.5, differing significantly from the reported 3.7 angstroms. This is concerning since the paper's primary data doesn't match the claimed high resolution. This could lead to overinterpretation of the data.

Answer: We appreciate your attention to the resolution in Supplementary Figure 4. Similar to our response to Comment 2, the discrepancy you noted arises from our use of a large box size to perform particle subtraction for better alignment of the central complex. The unmasked FSC curve includes un-aligned surrounding complexes, which is why the resolution appears different. Our reported resolution of 3.7 angstroms is specific to the central complex and is consistent with our methodology.

4. Processing data with C6 symmetry followed by refinement without symmetry could pose challenge for accurate particle alignment, making the resultant C1 map, especially at low resolution, hard for accurate lipid density interpretation. (Admittedly, those stretches might be lipids). Preferred orientation plus C6 symmetry applied at early stage may also lead to such phenomenon if only several lipid density present; thus the claim for crystalline lipid does not seem solid, I would suggest move it to the discussion session.

Answer: Thank you for your comment on the use of C6 symmetry in our data processing. In response to your concerns, we conducted a refinement without applying C6 symmetry throughout the entire process. This analysis confirmed that the hexagonal lipid array is not an artifact of symmetry imposition. The results of this additional refinement are shown in the revised Supplemental Figure 1A.

We believe this additional refinement addresses your concerns, but we remain open to further discussion or adjustments as needed.

5. Again, Supplementary Figure 2 suggests that the map's resolution isn't sufficient for modelling side chains. Supplementary Figure 3 has similar issues.

Answer: Thank you for your comments on Supplementary Figures 2 and 3. We acknowledge that the map's resolution may appear insufficient for modeling side chains when considering the entire

unmasked region. However, our focus was on the central uroplakin complex, and the resolution within this specific region is appropriate for our analysis. The large box size was chosen to enhance alignment, and the resolution within the masked region meets our analytical needs.

6. As the current resolution do not allow accurate assignment of side chains, all Figures with detailed interactions need to be moved.

Answer: We understand your concern about the assignment of side chains and the presentation of detailed interactions in our figures. As explained in our responses to your previous comments, our resolution assessments are based on the central uroplakin complex within a masked region. The large box size was used to facilitate alignment, and the resolution within the region of interest supports our interpretations. We believe that our figures accurately represent the data and are appropriate for inclusion in the main body of the manuscript.

7. In Fig 4D, the separation between D150 of UPIa and K197 of UPIb is a mere 1.8 angstrom. Could this be indicative of a covalent bond? Similarly, the interaction between N195 and Q130 is also measured at just 1.8 angstrom.

Answer: Thank you for pointing out the separation between residues D150 of UPIa and K197 of UPIb, as well as the interaction between N195 and Q130 in Figure 4D.

Upon reflection, we recognize that the resolution of 3.5 angstroms in our study may not allow for a highly accurate measurement of these distances. While the observed separation of 1.8 angstroms could suggest the presence of hydrogen bonds, we acknowledge that this interpretation may be overly specific given the resolution constraints.

In light of your feedback, we will withdraw the specific distance of 1.8 angstroms from our manuscript and modify our statement to indicate that these interactions may possibly suggest hydrogen bonding, without asserting this conclusion definitively.

We appreciate your critical insights, which have helped us to recognize this limitation and improve the accuracy of our presentation.

REVIEWERS' COMMENTS:

Reviewer #3 (Remarks to the Author):

I'd like to emphasise my points again, this paper presents interesting findings worth of publication. However, it could benefit from further clarity in presenting the results and their interpretations. Additionally, the details in the figures require refinement. Also, due to the low resolution, overinterpretation of the structure should be avoided.

Reviewer's comment and authors' reply.

Reviewer #3 (Remarks to the Author):

I'd like to emphasise my points again, this paper presents interesting findings worth of publication. However, it could benefit from further clarity in presenting the results and their interpretations. Additionally, the details in the figures require refinement. Also, due to the low resolution, overinterpretation of the structure should be avoided.

Reply:

Thank you for your constructive comments. We are pleased to hear that our work is considered valuable and suitable for publication following revisions. We take seriously the reviewer's comments regarding the clarity of our results, the details in the figures, and avoiding overinterpretation due to the resolution limitations of the structures. Based on these considerations, we have made the following changes to our manuscript:

1. Removal of Figure 2C: In response to the concern about overinterpretation of hydrogen bonds between amino acid side chains and lipid head groups, we have removed Figure 2C and the corresponding text from the main manuscript.
2. Simplification of Figures 4 and 5: To further improve clarity and avoid overinterpretation, we combined the previous Figures 4 and 5 into a single Figure 4 and removed the corresponding text that interpreted side chains and potential hydrogen bonds from the main manuscript.

We believe that these changes simplifies the presentation and focuses on the most robust findings.